# Machine Learning Improves the Prediction Rate of Non-Curative Resection of Endoscopic Submucosal Dissection in Patients with Early Gastric Cancer

**DOI:** 10.3390/cancers14153742

**Published:** 2022-07-31

**Authors:** Hae-Ryong Yun, Cheal Wung Huh, Da Hyun Jung, Gyubok Lee, Nak-Hoon Son, Jie-Hyun Kim, Young Hoon Youn, Jun Chul Park, Sung Kwan Shin, Sang Kil Lee, Yong Chan Lee

**Affiliations:** 1Department of Internal Medicine, Yongin Severance Hospital, Yonsei University College of Medicine, Seoul 16995, Korea; siberian82@yuhs.ac; 2Department of Internal Medicine, Severance Hospital, Yonsei University College of Medicine, Seoul 16995, Korea; junchul75@yuhs.ac (J.C.P.); kaarma@yuhs.ac (S.K.S.); sklee@yuhs.ac (S.K.L.); leeyc@yuhs.ac (Y.C.L.); 3Graduate School of AI, College of Engineering, Korea Advanced Institute of Science and Technology (KAIST), Daejeon 34141, Korea; gyubok.lee@kaist.ac.kr; 4Department of Statistics, Keimyung University, Daegu 42601, Korea; nhson@ms.kmu.ac.kr; 5Department of Internal Medicine, Gangnam Severance Hospital, Yonsei University College of Medicine, Seoul 16995, Korea; otilia94@yuhs.ac (J.-H.K.); dryoun@yuhs.ac (Y.H.Y.)

**Keywords:** early gastric cancer, non-curative resection, machine learning, prediction

## Abstract

**Simple Summary:**

Endoscopic submucosal dissection (ESD) is accepted as a standard treatment for early gastric cancer (EGC). Non-curative resection (NCR) of EGC after ESD can increase the burden of additional treatment and medical expenses. Thus, we aimed to develop a machine-learning (ML)-based NCR prediction model for EGC prior to ESD. We obtained data from 4927 patients with EGC who underwent ESD between January 2006 and February 2020. Seven ML-based NCR prediction models were developed using ten clinicopathological characteristics. The performance of NCR prediction was highest in the XGBoost model (AUROC, 0.851; 95% confidence interval, 0.837–0.864). Our ML model improved the ability to predict NCR of ESD in patients with EGC. This ML model can provide useful information for decision-making regarding the appropriate treatment of EGC before ESD.

**Abstract:**

Non-curative resection (NCR) of early gastric cancer (EGC) after endoscopic submucosal dissection (ESD) can increase the burden of additional treatment and medical expenses. We aimed to develop a machine-learning (ML)-based NCR prediction model for EGC prior to ESD. We obtained data from 4927 patients with EGC who underwent ESD between January 2006 and February 2020. Ten clinicopathological characteristics were selected using extreme gradient boosting (XGBoost) and were used to develop a ML-based model. Dataset was divided into the training and internal validation sets and verified using an external validation set. Sensitivity, specificity, and area under the receiver operating characteristic curve (AUROC) were evaluated. The performance of each model was compared by using the Delong test. A total of 1100 (22.1%) patients were identified as being treated non-curatively with ESD. Seven ML-based NCR prediction models were developed. The performance of NCR prediction was highest in the XGBoost model (AUROC, 0.851; 95% confidence interval, 0.837–0.864). When we compared the prediction performance by the Delong test, XGBoost (*p* = 0.02) and support vector machine (*p* = 0.02) models showed a significantly higher performance among the NCR prediction models. We developed an ML model capable of accurately predicting the NCR of EGC before ESD. This ML model can provide useful information for decision-making regarding the appropriate treatment of EGC before ESD.

## 1. Introduction

Endoscopic submucosal dissection (ESD) is accepted as a standard treatment for early gastric cancer (EGC) that satisfies prespecified criteria. The several guidelines defined curative resection as an en bloc resection of EGC that showed negative horizontal and vertical margins and met expanded indication. Any ESD that does not satisfy these criteria is considered a non-curative resection (NCR) [1,2].

The overall incidence of NCR after ESD for EGC is about 15–20% [3,4,5]. Additional treatment is required following NCR because of the risk of local recurrence or lymph node metastasis. Therefore, clinicians should strive to select eligible patients with EGC for ESD to avoid additional treatment after NCR. However, even if EGC is regarded as a lesion within the expanded inclusion criteria before ESD, many patients still require additional treatments following ESD because of NCR [3,4,5,6]. Therefore, several studies have developed models to predict NCR and reduce the incidence of NCR after ESD [7,8,9,10,11]. At our center, we developed a risk-scoring system (RSS) to predict NCR using large prospectively collected cohort data [7]. The RSS showed good performance in predicting NCR after ESD in patients with EGC. However, the limitations of this RSS were that external validation was not performed and was conducted at a single center.

Recently, artificial intelligence has been applied in clinical practice because of its high precision, accuracy, speed, and low error rate compared to humans [12,13,14]. Machine learning (ML) is a discipline that uses computational modeling to learn from data, meaning that performance in executing a specific task improves with experience. Thus, ML models may improve the risk stratification ability provided by existing clinical risk factors used to develop a prediction model. However, previous ML models that were used to predict NCR after ESD in patients with EGC had limitations, such as small sample sizes, lack of external validation, and/or absence of head-to-head comparisons, among prediction models [8,15]. Thus, we developed ML-based models for the prediction of NCR prior to ESD in patients with EGC and compared the performance of the ML-based models, including our previous RSS.

## 2. Materials and Methods

### 2.1. Data Collection and Study Population

We retrospectively reviewed data of patients with EGC who underwent ESD at Severance Hospital between January 2006 and February 2020 and at Gangnam Severance Hospital between April 2012 and November 2018. Among 14,004 ESD patients, we excluded 101 patients without endoscopy, tumor, and histology results. In addition, we further excluded 8976 patients with low grade dysplasia (*n* = 5410), high grade dysplasia (*n* = 2694), subepithelial tumors (*n* = 315), and others (*n* = 557). Finally, 4927 patients who underwent ESD (4396 at Severance Hospital and 531 at Gangnam Severance Hospital) were included in the study (Figure 1). We analyzed the patients’ age, sex, antithrombotic agent, tumor size and location, tumor histology, multiplicity of tumors, and various endoscopic findings of EGCs. The clinicopathological characteristics of these patients, including the endoscopic findings, were obtained through a retrospective review of medical records and compared between the groups.

The entire dataset consisted of data from two tertiary general hospitals (Severance Hospital and Gangnam Severance Hospital). The dataset was partitioned into training, internal validation, and external validation sets according to the participating center to ensure the generalizability of the algorithm. Consequently, data from 4396 patients from Severance Hospital were used for training (90%) and internal validation (10%). From Gangnam Severance Hospital, data of 531 patients were used for external validation (100%). To develop the NCR prediction algorithm, each variable for the prediction models was examined according to clinical relevance and importance scores using a gradient boosting (XGBoost) model. The details of the feature importance plot are shown in Appendix A. We then categorized the clinicopathological characteristics into the following four categories: (1) demographics (age, sex, and use of antithrombotics); (2) endoscopy (gross appearance and detailed findings); (3) tumors (size, multiplicity, and location with respect to the long and short axis); and (4) histology (Table 1). 

This study was conducted in accordance with the principles of the Declaration of Helsinki, and the study protocol was approved by the Institutional Review Board of the Yonsei University Health System Clinical Trial Center (2022-0236-001). As the clinical data used in the model development were collected retrospectively, the requirement for obtaining informed consent was waived. 

### 2.2. Definitions

The expanded indications for curative ESD are en bloc resection, negative horizontal and vertical margins, no lymphovascular invasion, and one of the following: (1) tumors > 2 cm, differentiated type, mucosa, and ulcer (−); (2) tumor ≤ 3 cm, differentiated type, mucosa, and ulcer (+); (3) tumor ≤ 2 cm, undifferentiated type, mucosa, and ulcer (−); or (4) tumor ≤ 3 cm, differentiated type and submucosa1 (SM1, <500 µm from the muscularis mucosa) [1,2]. En bloc resection was defined as the removal of gastric lesions in a single piece without fragmentation. Consequently, NCR was defined as a resection that did not satisfy any of the above criteria. 

### 2.3. ESD Technique 

All ESD procedures were performed in hospitalized patients. Immediately before the procedure, midazolam hydrochloride or propofol was administered intravenously for sedation. All patients were examined using a video endoscope with or without a water-jet function (GIF-HQ290, GIF-Q260, and GIF-H260; Olympus, Tokyo, Japan) while lying in the left lateral decubitus position. After the endoscopic examination of the gastric lesions, the area surrounding each lesion was marked using argon plasma coagulation (VIO 300D; ERBE, Tübingen, Germany). A saline solution containing epinephrine (0.01 mg/mL) and 0.8% indigo carmine was injected into the submucosal layer to elevate the lesion from the muscle layer. A dual knife (KD-650Q; Olympus, Tokyo, Japan) or insulated-tip knife (KD-610L; Olympus Optical, Tokyo, Japan) was used to make a circumferential incision and dissection. Hemoclips or hemostatic forceps were used to control the bleeding or exposed vessels. All patients underwent chest and abdominal radiography immediately after ESD and on the first day after ESD to detect adverse outcomes, such as pneumonia or perforation. After ESD, all patients were administered proton pump inhibitors for 4–8 weeks.

### 2.4. Gross and Histopathologic Evaluation 

The endoscopic findings of EGC were classified according to the criteria of the Japanese Research Society for Gastric Cancer [16]. All specimens were sectioned at 2-mm intervals, centered on the part of the lesion closest to the margin and the site of the deepest invasion. Slides stained using hematoxylin and eosin were used for general evaluation. Tumor size, invasion depth, lymphatic and vascular involvement, and tumor involvement at lateral and vertical margins were assessed histologically.

### 2.5. Prediction Models

We developed the following seven ML models: logistic regression, support vector machine, k-nearest neighbors, naive bayes, extreme gradient boosting (XGBoost), random forest, and multilayer perceptron. Ten-fold cross-validation was conducted on the internal validation group (90% for training and 10% for interval validation). The final performance of the ML models was the average result of the 10-fold cross-validation. A grid search was utilized to optimize the hyperparameters of each ML model. After the models were constructed, each prediction model was further evaluated in the external validation group. The detailed architecture of the ML-based NCR prediction model is shown in Figure 2. All ML models were implemented using the scikit-learn package version 0.24.2 using Python 3.7.6 throughout the experiments.

### 2.6. Statistical Analysis

Continuous variables are presented as means and standard deviations. Categorical variables were presented as numbers and percentages. To compare the difference between internal and external datasets, independent two-sample *t* tests and chi-square tests were used for continuous and categorical variables, respectively. The Kruskal–Wallis test was used for data with skewed distribution. The performance of the NCR prediction models was evaluated using the sensitivity, specificity, precision, F1 score, area under the receiver operating characteristics curve (AUROC), and area under the precision-recall curves as follows:$$\mathrm{Sensitivity}\;\left(\mathrm{Recall}\right):\;\frac{\mathrm{TP}}{\mathrm{TP}+\mathrm{FN}}\;\mathrm{Specificity}:\;\frac{\mathrm{TN}}{\mathrm{TN}+\mathrm{FP}}$$
$$\mathrm{Precision}:\;\frac{\mathrm{TP}}{\mathrm{TP}+\mathrm{FP}}\;F1\;\mathrm{score}:\;\frac{2\times \mathrm{Precision}\times \mathrm{Recall}}{\mathrm{Precision}+\mathrm{Recall}}$$
where TP, true positive; FN, false negative; TN, true negative; and FP, false positive.

The curves were plotted by varying the thresholds, and the areas under the curves were compared using the DeLong test. The interpretation of the AUROC and the area under the precision-recall curves was achieved by comparing the values among the models. All the performance indices were measured using an external validation set. Statistical analysis was performed using STATA, version 16.1 (Stata Corporation, College Station, TX, USA). Statistical significance was set at *p* < 0.05.

## 3. Results

### 3.1. Baseline Characteristics of the Patients

The baseline characteristics are shown in Table 2. The mean age of the study population was 64.4 years, and 3620 (73.5%) patients were men. Tumors were commonly located in the lower portion and lesser curvature in both datasets. The tumor size was significantly larger in the external dataset compared with the internal dataset (12.4 ± 8.4 mm vs. 20.0 ± 12.7 mm, *p* < 0.001). Tumor histology and endoscopic findings, including ulcer (*p* < 0.001), a fusion of fold, interruption or smooth tapering of the fold (*p* < 0.001), erythema (*p* < 0.001), whitish scar or atrophy (*p* = 0.002), nodularity or elevated (*p* < 0.001), and spontaneous bleeding (*p* < 0.001), were significantly different across the dataset. In addition, the proportion of undifferentiated type histology, such as poorly differentiated adenocarcinoma (113 [2.6%] vs. 32 [6.0%], *p* < 0.001) and signet-ring cell type (210 [4.8%] vs. 40 [7.5%], *p* < 0.001) was significantly higher in the external dataset than in the internal dataset.

### 3.2. Performance of the ML Model for Prediction of NCR

Among 4972 patients, the rate of NCR after ESD was 22.1%. The NCR rate was significantly higher in the internal dataset compared with the external dataset (24.2% vs. 6.4%). Detailed clinicopathological features according to the curative resection and NCR groups are summarized in Appendix A. In the internal dataset, among seven ML models, the AUROC predicting NCR was highest in the XGBoost model (0.851; 95% confidence interval [CI], 0.837–0.864), followed by logistic regression (0.840; 95% CI, 0.825–0.854), multilayer perceptron (0.837; 95% CI, 0.823–0.850), random forest (0.812; 95% CI, 0.797–0.827), k-nearest neighbors (0.807; 95% CI, 0.792–0.822), and naive Bayes (0.799; 95% CI, 0.783–0.815) (Figure 3A, Appendix A). Among the ML models, the XGBoost model outperformed the prediction rate for NCR. In the external dataset, the XGBoost model also showed superior performance among other ML methods with an AUROC of 0.710 (95% CI, 0.612–0.803), followed by logistic regression (0.693; 95% CI, 0.610–0.773), support vector (0.693; 95% CI, 0.613–0.769), and multilayer perceptron (0.691; 95% CI, 0.603–0.771) (Figure 3B, Appendix A). The sensitivity, specificity, precision, F1 score, and AUROC of the seven ML models are summarized in Table 3. Similarly, F1 scores were consistently higher in the XGBoost model as compared with other models, irrespective of the threshold (Appendix A).

The decision curve analyses showed that the net benefit of the XGBoost model was greater than other ML models, and the naive Bayes model had the lowest benefit when compared with other models (Appendix A).

In a previous study [7], we developed an RSS using logistic regression modeling in approximately 1600 patients with EGC who underwent ESD. When we applied this RSS to the internal dataset (training and internal validation sets), the AUROC of the previous RSS was 0.701 (95% CI, 0.683–0.720). When the previous RSS was applied to the external dataset that included 531 EGCs patients, the AUROC of the previous RSS model was 0.616 (95% CI, 0.516–0.719) (Appendix A). The XGBoost model outperformed the previous RSS (0.851 vs. 0.701, *p* < 0.001) up to a difference of 0.150 in the internal dataset. When we compared the prediction performance using the Delong test, the ML model, except for the support vector, showed a significantly higher performance compared with the previous RSS developed without ML in our center (all *p* < 0.001) (Table 3). However, when we compared the performance of AUROC between the seven ML models, they were not significantly differ (Appendix A).

## 4. Discussion

Our research identified that an XGBoost ML model derived from two large cohorts significantly improved the prediction of NCR of ESD in EGC in internal and external validations. Thus, this ML model improved the ability to predict NCR of ESD in patients with EGC. ML is defined as a computer-aided prediction method, with the most significant benefit being an increase in forecast accuracy for NCR prior to ESD.

ESD is widely performed for EGC and has favorable outcomes [17]. However, the number of cases requiring additional treatment due to NCR after ESD has also increased [5,18]. According to a recent study, the rate of additional surgery due to NCR after ESD was reported to be 4.3% [18]. However, additional gastrectomy can lead to a waste of medical resources and expenses and increase the risk of adverse events. Therefore, an accurate prediction of NCR of EGC prior to ESD is beneficial.

Various risk factors associated with NCR of ESD for EGC are reported in several studies [7,8,9,10,11]. Among several studies, our group developed an RSS comprising several endoscopic findings [7]. Logistic regression modeling revealed that a large tumor size (≥20 mm), tumor location in the upper body in the stomach, presence of an ulcer, a fusion of gastric folds, absence of mucosal nodularity, spontaneous bleeding, and undifferentiated tumor histology on biopsy were associated with NCR of ESD. The risk score points were assigned for seven variables based on the beta coefficient divided by the absolute value of the smallest coefficient. Each variable was scored as one or two points, and the total score was calculated as the sum of the components. The final score ranged from 0 to 7. This RSS showed an acceptable discriminatory performance in internal validation (AUROC, 0.7004). However, the RSS had no external validation. After this study, although several prediction models for NCR of ESD were proposed, there were limitations in that the sample size was small or only internal validation was performed [8,9,10,11].

However, several studies have attempted to improve the curative resection rate by predicting the depth of cancer accurately or the margin of tumor using endoscopic ultrasound (EUS) or magnifying endoscopy with narrow-band imaging (M-NBI) [19,20,21,22,23,24]. However, EUS and M-NBI also have limitations owing to inter- and intra-observer variabilities and lesion characteristics, such as cancer location or gross appearance. To overcome these limitations, ML is emerging as an alternative method to improve prediction performance [25].

An important finding of our study is the presentation of the determination reason or process of the ML model through explainable artificial intelligence analysis. Notably, there is a compromise between accuracy and interpretability in the ML classification model. Although the ML approach exhibited high degrees of accuracy based on complex calculations, it is characterized by low interpretability (artificial intelligence is more generally characterized as being of a “black-box nature”). The XGBoost classifier used parallel tree boosting analysis to provide highly efficient and accurate predictions [26,27]. In our study, the ML model significantly increased the AUROC from 0.701 with our previous RSS to 0.851 in the internal dataset, even though when we applied our previous RSS to the external validation set, the prediction rate of our previous RSS was 0.616. However, the ML model outperformed our previous RSS in the external validation dataset. The strength of our study was the large sample size from two tertiary general hospitals and internal and external validations were performed. Importantly, our ML model significantly outperformed previous RSS using logistic regression modeling without ML in both the internal and external validation datasets. Using this ML model could significantly reduce the rate of NCR, avoid using additional medical resources, and increase total costs.

Although this study established and rigorously validated the predictive performance of the designed ML model, it had several limitations. First, there was some discrepancy in the validation performance between the internal and external validation sets. Second, we used a prospectively collected database, and the analysis was retrospective. Thus, prospective validation should be performed to validate ML models. Third, although the ML model outperformed the previous RSS, the prediction performance of ML may have been affected by the model design, including hyperparameters. In addition, undetermined significant variables are not identified or analyzable before ESD, which limits the predictive power of the models. Fourth, there was no comparison of the performance of predictive models other than ML models, such as generative adversarial network models. Fifth, our algorithm is a black-box model; thus, we do not have information or an understanding of how the algorithm operates internally. Finally, our ML model was based on a Korean population; thus, our ML model may not be generalizable to other ethnic groups. Nevertheless, our ML model can significantly improve the prediction rate of NCR in both internal and external datasets. Individualized prediction using pre-ESD variables based on the ML model can help physicians’ decision-making processes. Thus, if our ML model is applied to patients with EGC who will undergo ESD, physicians can determine the prediction rate of NCR. Therefore, if the NCR rate is high, physicians may discuss options other than ESD, such as surgery, with patients. However, a prospective study is required to validate this model.

## 5. Conclusions

In conclusion, we developed a ML model capable of accurately predicting NCR of EGC before ESD by considering the demographic and endoscopic characteristics of the lesions. Our ML model outperformed the previous RSS for NCR of ESD in patients with EGC. This ML model can provide useful information for decision-making regarding the appropriate treatment of EGC before ESD.

## Figures and Tables

**Figure 1 cancers-14-03742-f001:**
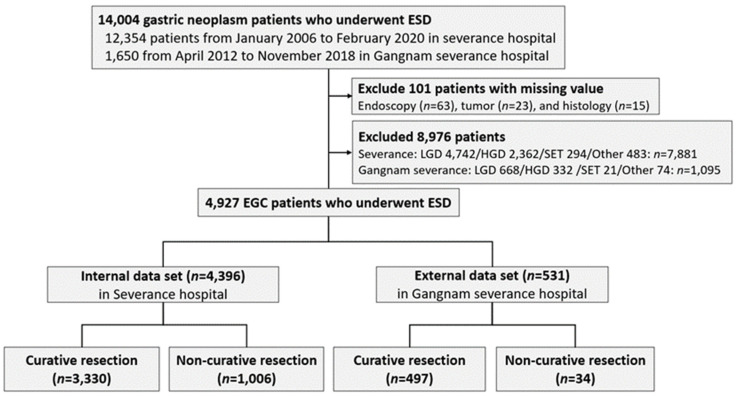
Flow diagram of study design. Abbreviations used are EGC, early gastric cancer; ESD, endoscopic submucosal dissection; LGD, low-grade dysplasia; HGD, high-grade dysplasia; and SET, subepithelial tumor.

**Figure 2 cancers-14-03742-f002:**
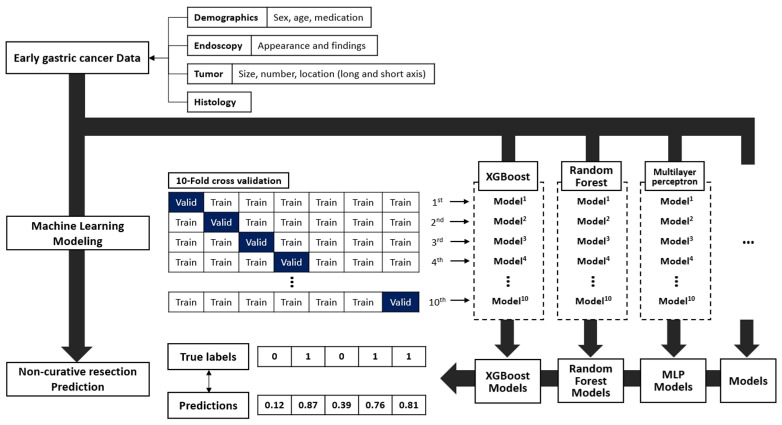
Development of non-curative resection after endoscopic submucosal dissection prediction model.

**Figure 3 cancers-14-03742-f003:**
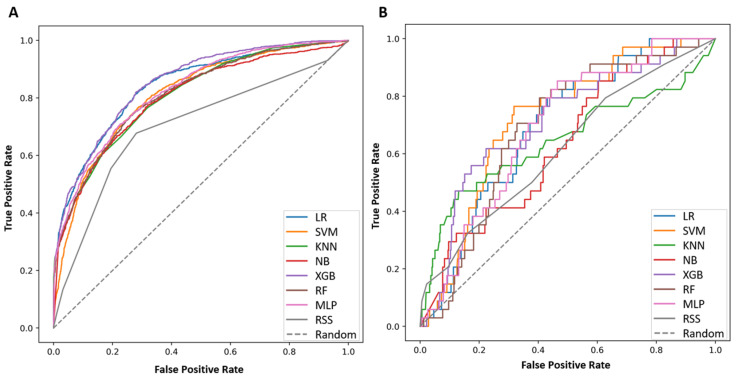
The area of the receiver operating characteristics for prediction of non-curative resection after endoscopic submucosal dissection. (**A**) Internal data set, (**B**) External data set. Abbreviation: RSS, risk-scoring system; LR, logistic regression; SVM, support vector machine; KNN, k-nearest neighbors; NB, naive bayes; XGB, extreme gradient boosting; RF, random forest; MLP, multilayer perceptron.

**Table 1 cancers-14-03742-t001:** Clinical variables used to build machine-learning models.

Category		Variables
Demographics		Sex, Age, Antithrombotics
Endoscopy	Appearance	Elevated, Flat, Depressed
	Finding	Ulcer, Fold, Erythema, Exudate, Whitish or atrophy, Nodularity or elevated, Spontaneous bleeding
Tumor	Size	Size (mm)
	Number	1, 2, or >2
	Location (long axis)	Upper, Middle, Lower
	Location (short axis)	Anterior wall, Posterior wall, Greater curvature, Lesser curvature
Histology		Adenocarcinoma well-differentiated, Adenocarcinoma moderate-differentiated, Adenocarcinoma poorly differentiated, Signet-ring cell, Others (Mucinous, Carcinoma in situ, Squamous cell type, etc)

**Table 2 cancers-14-03742-t002:** Baseline characteristics of the internal and external data set.

Variables	Overall (*n* = 4927)	Internal Data Set (*n* = 4396)	External Data Set (*n* = 531)	*p*-Value
Demographics				
Age, years	64.4 ± 10.2	64.7 ± 10.1	62.4 ± 11.2	<0.001
Male	3620 (73.5)	3240 (73.7)	151 (71.6)	0.29
Medications				
Antithrombotics	923 (18.7)	830 (18.9)	93 (17.5)	0.45
Histology				
AWD	1565 (31.8)	1425 (32.4)	140 (26.4)	<0.001
AMD	1228 (24.9)	1109 (25.2)	119 (22.4)	0.16
APD	145 (2.9)	113 (2.6)	32 (6.0)	<0.001
SRC	250 (5.1)	210 (4.8)	40 (7.5)	<0.001
Other (Mucinous, CIS, SCC)	1739 (35.2)	1539 (35.0)	200 (37.7)	0.23
Multiple lesions				
1	4280 (86.9)	3785 (86.1)	495 (93.2)	0.01
2	561 (11.4)	532 (12.1)	29 (5.5)	<0.001
>2	647 (13.1)	611 (13.9)	36 (6.8)	<0.001
Tumor location (long axis)				
Upper	495 (10.0)	441 (10.0)	54 (10.2)	0.92
Mid	1681 (34.1)	1463 (33.3)	218 (41.1)	<0.001
Lower	2635 (53.5)	2369 (53.9)	266 (50.1)	0.10
Tumor location (short axis)				
AW	1050 (21.3)	927 (21.1)	123 (23.2)	0.27
PW	1189 (24.1)	1070 (24.3)	119 (22.4)	0.33
LC	1838 (37.3)	1607 (36.6)	231 (43.5)	<0.001
GC	1023 (20.8)	904 (20.6)	119 (22.4)	0.32
Tumor size (mm)	13.1 ± 9.2	12.4 ± 8.4	20.0 ± 12.7	<0.001
Endoscopic appearance				
Elevated	3253 (66.0)	2976 (67.7)	277 (52.2)	<0.001
Flat	1341 (27.2)	1124 (25.6)	217 (40.9)	<0.001
Depressed	2302 (46.7)	2029 (46.2)	273 (51.4)	0.02
Endoscopic finding				
Ulcer	274 (5.6)	225 (5.1)	49 (9.2)	<0.001
Fusion of fold, interruption, or smooth tapering of fold	104 (2.1)	70 (1.6)	34 (6.4)	<0.001
Erythema	795 (16.1)	534 (12.1)	261 (49.2)	<0.001
Exudate	210 (4.3)	102 (2.3)	107 (20.2)	<0.001
Whitish scar or atrophy	269 (5.5)	225 (5.1)	44 (8.3)	0.002
Nodularity or elevated	863 (17.5)	596 (13.6)	267 (50.3)	<0.001
Spontaneous bleeding	60 (1.2)	38 (0.9)	22 (4.1)	<0.001

Note: Values for categorical variables are given as a number (percentage); values for continuous variables are given as mean (standard deviation). Abbreviations: AMD, adenocarcinoma moderate-differentiated; AWD, adenocarcinoma well-differentiated; APD, adenocarcinoma poorly differentiated; SRC, signet-ring cell; CIS, carcinoma in situ; SCC, squamous cell carcinoma; AW, anterior wall; PW, posterior wall; LC, lesser curvature; GC, greater curvature.

**Table 3 cancers-14-03742-t003:** Performance of the non-curative resection prediction model for the seven machine-learning models used in this study.

Risk Score	Precision	F1 Score	AUPRC (95%CI)	Sensitivity	Specificity	AUROC (95%CI)	*p*-Value ^a^
Internal data							
RSS	0.636	0.777	0.463 (0.449–0.478)	0.998	0.008	0.701 (0.683–0.720)	
LR	0.735	0.547	0.691 (0.677–0.705)	0.788	0.721	0.840 (0.825–0.854)	<0.001
SVM	0.700	0.460	0.596 (0.581–0.610)	0.827	0.618	0.667 (0.647–0.687)	<0.001
KNN	0.835	0.436	0.652 (0.637–0.665)	0.771	0.665	0.807 (0.792–0.822)	<0.001
NB	0.696	0.492	0.633 (0.619–0.647)	0.946	0.380	0.799 (0.783–0.815)	<0.001
XGB	0.749	0.576	0.699 (0.685–0.713)	0.785	0.732	0.851 (0.837–0.864)	<0.001
RF	0.925	0.326	0.647 (0.633–0.661)	0.713	0.757	0.812 (0.797–0.827)	<0.001
MLP	0.718	0.527	0.676 (0.662–0.689)	0.722	0.752	0.837 (0.823–0.850)	<0.001
External data							
RSS	0.200	0.333	0.174 (0.163–0.186)	0.977	0.147	0.616 (0.516–0.719)	
LR	0.122	0.193	0.104 (0.095–0.113)	0.561	0.794	0.693 (0.610–0.773)	0.09
SVM	0.099	0.133	0.113 (0.104–0.122)	0.563	0.794	0.693 (0.613–0.769)	0.02
KNN	0.202	0.148	0.169 (0.159–0.181)	0.829	0.470	0.645 (0.523–0.762)	0.69
NB	0.096	0.147	0.151 (0.141–0.162)	0.776	0.411	0.631 (0.540–0.722)	0.74
XGB	0.187	0.274	0.125 (0.116–0.135)	0.587	0.735	0.710 (0.612–0.803)	0.02
RF	0.031	0.030	0.099 (0.090–0.108)	0.394	0.911	0.688 (0.604–0.769)	0.12
MLP	0.126	0.188	0.105 (0.096–0.114)	0.551	0.823	0.691 (0.603–0.771)	0.06

^a^ Compared with the area under the receiver, operating characteristics of the score-based non-curative resection prediction model used by the Delong test. Abbreviation: AUPRC, the area under the precision-recall curve; AUROC, the area under the receiver operating characteristics curve; CI, confidence interval; RSS, risk-scoring system; LR, logistic regression; SVM, support vector machine; KNN, k-nearest neighbors; NB, naive bayes; XGB, extreme gradient boosting; RF, random forest; and MLP, multilayer perceptron.

## Data Availability

The data used in this study are only available from Severance Hospital, Gangnam Severance Hospital, or Yongin Severance Hospital upon reasonable request.

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
