# Peer review of "Machine Learning Improves the Prediction Rate of Non-Curative Resection of Endoscopic Submucosal Dissection in Patients with Early Gastric Cancer"

_cancers, 2022, doi:10.3390/cancers14153742_

Round 1

Reviewer 1 Report

The paper is well written and analyze a topic of current interest. The study is relevant, however, I am not sure of the clinical utility of the research results presented.

1. The introduction is short and succinct, which is a big advantage. However, please also refer to the ESGE guidelines. This will raise the international status of the study.  

2. Line 82-83 says "We also excluded 101 patients who had more than 50% of missing data." Does this mean that some patients had missing data? How high was the missing data percentage? Which data were missing most often? What impact did this have on the results of the ML models?  

3. One of the factors used in ML models is the histopathological diagnosis of the lesion. Is it known in every case before the ESD procedure? 

4. One dataset comes from one Severance Hospital and another from Gangnam Severance Hospital. Why wasn't all the data anonymised, mixed and then split into a training, validation and test set? In the 'Discussion' section, the discrepancies between the data sets are mentioned. Please describe what impact this has on the results and clinical utility of the ML models.

5. Please include information on which statistical tests were used in Table 2.

6. Please remove lines 242-245 from the "Discussion" section.

7. The precision values in Table 3 for the external dataset and the precision-recall curve in Supplementary Figure 2B seems to be statistically insignificant. Have these results been tested for statistical significance? I suggest comparing the AUPCR values with each other and with a random model using, for example, this method: https://www.medcalc.org/manual/precision-recall.php

8. Please also perform Delong tests between all ML models and include the results in an additional Table in Supplementary Materials.

Reviewer 2 Report

The authors tried to develop a machine-learning (ML)-based NCR prediction model for EGC prior to ESD. And the authors developed an ML model capable of accurately predicting the NCR of EGC before ESD.

Please clarify these points described below.

The authors used several variables from medical records. How do you select these parameters for this prediction model?

In the definition section in Methods, the authors described that NCR was defined as a resection that did not satisfy any of the above criteria. And used criteria were the indications for gastric ESD. That means authors’ NCR definitions included two or more different categories about ESD failure. For developing a NCR prediction model, the prediction target should be simpler. How do you think about the definitions of NCR in this point?

As external set, the authors used the data from Gangnam Severance Hospital. This external data set was quite different from original data set. Why the authors used so different data set for this analysis?

Round 2

Reviewer 1 Report

Thank you very much for the responses and improving the manuscript. I appreciate the effort and work put into the study, however, considering all the statistical results the ML model obtained is not sufficient to accurately predict every NCR of EGC before ESD.

1. Please add the information that the results indicate the significant potential of ML models in this field, but still a significant part of the variables are not identified or analyzable before ESD, which limits the predictive power of the models.

2. Please also include an interpretation of the AUCROC score of the best ML model. According to the Mandrekar article, the AUCROC result of external data is only accetable.   

Mandrekar JN. Receiver operating characteristic curve in diagnostic test assessment. J Thorac Oncol. 2010 Sep;5(9):1315-6. doi: 10.1097/JTO.0b013e3181ec173d. PMID: 20736804

Author Response

  1. Please add the information that the results indicate the significant potential of ML models in this field, but still a significant part of the variables are not identified or analyzable before ESD, which limits the predictive power of the models.

Thanks for the good suggestion. We add your suggestion in the limiations.

  1. Please also include an interpretation of the AUCROC score of the best ML model. According to the Mandrekar article, the AUCROC result of external data is only accetable

Thanks for the good suggestion. We had already used AUROC to evaluate all the results and interpret them in the manuscript. In order to confirm robustness in the external validation set, the results were also presented using the AUROC method. The statistically basic rules have been faithfully applied to this study and presented. Thank you for your attention.

Reviewer 2 Report

Thank you for your responses on my review.

Author Response

Thank you for your comment.